# Relationship between Environmental Conditions and Utilisation of Community-Based Mental Health Care: A Comparative Study before and during the COVID-19 Pandemic in Italy

**DOI:** 10.3390/ijerph21060661

**Published:** 2024-05-22

**Authors:** Eleonora Prina, Federico Tedeschi, Antonio Lasalvia, Damiano Salazzari, Sara Latini, Laura Rabbi, Federica Marando, Elaine van Rijn, Jan Wollgast, Enrico Pisoni, Bertrand Bessagnet, Maxime Beauchamp, Francesco Amaddeo

**Affiliations:** 1Department of Neurosciences, Biomedicine and Movement Science, Section of Psychiatry, University of Verona, 37134 Verona, Italy; federico.tedeschi@univr.it (F.T.); antonio.lasalvia@univr.it (A.L.); salazzari@gmail.com (D.S.); sara.latini@univr.it (S.L.); laura.rabbi@univr.it (L.R.); francesco.amaddeo@univr.it (F.A.); 2European Commission, Joint Research Centre (JRC), 21027 Ispra, Italy; federica.marando@ec.europa.eu (F.M.); elaine.van-rijn@ec.europa.eu (E.v.R.); jan.wollgast@ec.europa.eu (J.W.); enrico.pisoni@ec.europa.eu (E.P.); bertrand.bessagnet@ec.europa.eu (B.B.); 3IMT Atlantique, 655 Av. du Technopôle, 29280 Plouzané, France; maxime.beauchamp76@gmail.com

**Keywords:** community-based mental health care, COVID-19 pandemic, environmental conditions, Italy, register study, service utilisation

## Abstract

(1) Background: Lower socioeconomic status increases psychiatric service use, exacerbated during the COVID-19 pandemic by environmental stressors like air pollution and limited green spaces. This study aims to assess the influence of sociodemographic and environmental factors on mental health service utilisation. (2) Methods: This retrospective study uses an administrative database focusing on community mental health services in Northeast Italy. Spatial and temporal analyses were used to address space–time dependencies. (3) Results: Findings showed that sociodemographic factors like living in rented apartments and lower education levels predicted higher mental health service use. Environmental factors, such as elevated NO_2_ levels and, before the pandemic, lower solar radiation and tree cover, correlated with increased service utilisation. COVID-19 reduced most of the pre-existing differences associated with these factors across census blocks with a different composition of sociodemographic and environmental factors. (4) Conclusions: These findings contribute to a better understanding of the impact of the environment on public mental health.

## 1. Introduction

Populations living in urban areas are exposed to a wide range of environmental stressors, including air pollution, a lack of green and blue spaces, and high levels of traffic and noise, which in turn negatively affect their mental health status [1,2,3,4]. These could be some of the reasons why living in urban environments is associated with the development of psychiatric symptoms, such as anxiety and depression [5], among the population during the COVID-19 pandemic.

A cross-sectional study conducted in Scotland during the period of the COVID-19 pandemic, with the most stringent containment measures (e.g., lockdown), showed that psychological distress was worse if participants reported living in a deprived urban area, had no access to or sharing of residential outside space, or experienced fewer visits to green spaces [6]. On the other hand, opportunities exist to support mental health by encouraging the use of existing residential outside spaces, public green spaces, and soundscapes [7,8]. People with greater exposure to green–blue spaces are less likely to develop a common mental health disorder (such as depression), and this effect is modified by socio-economic deprivation [9]. In a recent study, urban nature was found to play a central role in creating more equitable, green, and liveable cities with active inhabitants [10]. This aligns with an umbrella review by Cuijpers et al. [11], which included 24 meta-analyses on the relationship between mental health and climate events, pollution, and green spaces. Among the included meta-analyses, only two specifically focused on green spaces, and the results suggested a small but significant association between exposure to green spaces and a reduction in mental health symptoms.

Besides green spaces, air pollution is also related to health. According to the Global Burden of Disease (GBD) report, in 2017, air pollution was responsible for up to 4.90 million deaths and 1.47 billion disability-adjusted life-years (DALYs) globally, with most of the burden related to cardiovascular and respiratory diseases [12]. More recently, the hazardous effects of air pollution on mental health problems, such as depression, have been suggested to have global public health implications [13,14]. Data from the World Health Organization (WHO) show that almost the entire global population (99%) breathes air that exceeds WHO guideline limits [15] and contains high levels of the common indices of air pollution. Among these, fine particulate matter (PM_2.5_) and nitrogen oxide (NO_2_) play an influential role in poor mental health conditions [16]. Depressive mood has been found to be positively associated with air pollution (see Rautio et al., 2018 for a review [17]).

Another environmental factor that can affect the urban population’s well-being is solar radiation. A recent epidemiological study highlighted that a reduced duration of sunshine has been associated with an increased risk of depression [18,19]. Moreover, solar radiation during the day before a suicide event has repeatedly been found to be significantly associated with an increased suicide risk itself [20,21,22].

Finally, there is a well-established link between lower socioeconomic status (SES) and an increased probability of utilising psychiatric services, particularly in socially deprived areas [23]. For instance, lower SES and limited education have been associated with worsened depression outcomes and higher mental health service utilisation [24,25]. The COVID-19 pandemic further exacerbated socioeconomic disparities, with reduced school hours and increased unemployment rates [26], and affected social connections, among other things. The COVID-19 pandemic posed a serious health risk for the population worldwide in terms of well-being and mental health, [27,28,29,30], especially for vulnerable groups such as people with chronic somatoform disorders, health care workers, gender minority individuals, people with a mental health condition, and COVID-19 patients [31,32,33,34,35]. Among these groups, as synthesised by a recent umbrella review by Bertolini et al. [36], there is strong evidence that people with pre-existing mental disorders suffered from worse physical and mental health outcomes due to the COVID-19 pandemic, as compared to the general population [37]. Several factors might come into play in determining such a negative outcome. People with severe mental illnesses, such as bipolar disorder and schizophrenia, more frequently live with a high body mass index (BMI), diabetes mellitus, generally limited exercise tolerance, and are also more likely to smoke and have substance abuse disorder [38]. This is also the case in Italy, one of the first nations among Western countries to be affected by the COVID-19 outbreak [39]. Despite this increase, both emergency psychiatric consultations and community mental health contacts were significantly reduced during the pandemic [27], at −23.3% and −33.9%, respectively [40,41]. The pandemic also brought the need for fast and flexible adaptations in health organisations to balance the increased demand with reduced interactions among patients and between patients and professionals. This inevitably affected those needing care the most [42,43,44].

The lockdown measures implemented to mitigate the spread of the coronavirus significantly influenced the levels of air pollution in Italy and numerous other countries. Notably, these restrictions led to a pronounced reduction in the concentrations of nitrogen dioxide, sulphur dioxide, and particulate matter (to a lesser extent) in urban areas [45]. These decreases in concentrations exceeded 50% for NO_2_ concentrations during the most stringent lockdowns; however, for secondary pollutants like ozone, the concentrations slightly increased due to the complex interplay between pollutants.

In the context of the pandemic, people living in urban environments were hypothesised to be extremely affected by the virus dissemination due to higher population density and higher concentrations of air pollutants, factors that could be associated with higher viral transmission and COVID-19 severity of infection [46].

In this study, we aim to examine, through a spatially explicit assessment, how sociodemographic and environmental conditions, including possible stressors and protective factors, affected patients’ use of mental health services in the city of Verona, and whether these relationships changed during a disruptive event, namely the COVID-19 pandemic. Our goal is to provide information on the possible impact of environmental factors on the utilisation of community-based mental health services, providing insights to policymakers on possible urban interventions to decrease the burden of disease.

## 2. Materials and Methods

We conducted a retrospective observational study employing an electronic administrative database that collects routine data from all individuals (≥18 years of age) seeking psychiatric and/or psychological care in the Verona Department of Mental Health (Northeast Italy) between 1 January 2019 and 30 June 2021. The catchment area of the Verona Department of Mental Health had a population of 926,497 inhabitants in 2019 (Eurostat data). All types of contacts (with psychiatrists, psychologists, nurses, psychiatric rehabilitation therapists, occupational therapists, and social workers) provided by mental health services were included. A comprehensive description of the study setting and clinical data sources can be found in a recently published article [41]. Our current study narrows its scope to the Verona municipality, diverging from the catchment area discussed in the previous publication and integrating environmental variables, an aspect previously omitted.

This study was conducted in accordance with the guidelines provided in the current version of the Declaration of Helsinki [47] and with the STROBE (STrengthening the Reporting of OBservational Studies in Epidemiology) Statement guidelines [48]. The STROBE Statement checklist is provided in Appendix A. The study protocol was approved by the local Ethics Committee of the Verona University Hospital Trust (3327CESC, Prot. 35819, 14 June 2021).

The outcome variable is the estimated rate of contacts with psychiatric and psychological services (per 10,000 inhabitants) of the adult population for each census block (CB) per week combination from January 2019 to June 2021. 

The dataset was divided into weeks, with the last ‘week’ of 2019 lasting 8 days, and the last week of 2020 lasting 9 days, while the last week of the first semester of 2021 lasted 6 days. This way, each contact could be ascribed to a given week (between 1 and 52 for 2019 and 2020 and between 1 and 26 for 2021) of a given year. A spatial autoregressive model for panel data was performed to account for the space–time structure of our data. As for space, a CB random effect (to allow for the dependence of errors of the same CB across time) and spatial autocorrelation of the errors (using an inverse distance spatial weighting matrix) were included in the model to control for possible omitted environmental variables and to take spatial dependence into account. The time structure of the data was accounted for by the inclusion of a week-level fixed effect (to account for seasonality) and the lagged value of the outcome (to allow for the persistence of the need for psychiatric help: CBs having had contacts in a given week are more likely to have contacts in the following week as well).

The weekly contact rate was divided by the number of working days to take into account that not all weeks have the same number of working days (due both to public holidays and to the longer length of the last week of each year). Therefore, the outcome variable is the estimated daily contact rate for the week. However, since non-working days could still have contacts (e.g., in case of emergencies) and weeks with public holidays could have fewer contacts than regular weeks, the proportion of working days in the week was inserted as a control variable in the regression.

### 2.1. Data Collection

Sociodemographic variables, including gender, age, citizenship, living situation, marital status, and the clinical variables of the patients with at least one contact in the study period were described through absolute numbers and percentages. In the case of multiple contacts, the one occurring first was considered. 

For sociodemographic variables, we categorised age as 18–24, 25–44, 45–64, and 65+; citizenship was grouped as Italian or foreigner; living situation included living alone, with family members, or in residential facilities/institution; marital status was divided into single, married, or separated/divorced/widowed. 

For clinical variables, we categorised diagnostic groups based on ICD-10 codes (WHO, 1992) as follows: schizophrenia and related disorders (codes F20 to F29), affective disorders (codes F30 to F39), neurotic or somatoform disorders (codes F40 to F48), personality disorders (codes F60 to F69), and other diagnoses (all other ICD-10 F and Z codes). Additionally, the ICD-10 codes were divided into two groups: the psychosis group (F20-F31 codes) and the non-psychosis group (all other ICD-10 F and Z codes) [49,50]. The trend of contacts of patients with psychosis (i.e., ICD-10 codes F20 to F31) was visually compared to that of the other patients through a graph. 

Environmental variables were included in our model as predictors. In particular, solar radiation (in kilowatts), PM_2.5_ concentration (PM_10_ concentrations were excluded due to multicollinearity with this variable), NO_2_ concentration, percentage of trees in a CB, share of green areas in a CB, and the presence of watercourses and large public green areas within 300 metres of the CB centroid were included. In the case of time-varying data (solar radiation and PM_2.5_ and NO_2_ concentrations), lagged variables were included (on the grounds that the decision to use psychiatric services may be affected by the previous week’s values of environmental variables) of solar radiation, and of the percentage of days in the previous week where specific thresholds (25 µg m^−3^ for PM_2.5_, and 50 µg m^−3^ for NO_2_) were exceeded. 

The following sociodemographic variables were included in the model as potential predictors: the low-schooling index (percentage of inhabitants aged 6 or more with primary school or a lower educational level), the unemployment rate (given by the ratio between the unemployed, i.e., the population aged 15–74 seeking a job, and the sum of employed and unemployed), and the percentage of households living in rented accommodation. Details on data sources and the construction of the variables are reported in Appendix A.

### 2.2. Model Estimating Air Pollutant Concentrations

PM_2.5_ and NO_2_ concentration maps were derived from the Copernicus Atmospheric Monitoring Service (CAMS) [51] with an additional downscaling procedure. We used a Kriging-based interpolation of the residuals [52] to get high-resolution concentrations at ground level. A spatial trend, denoted as the deterministic part of the model, is fitted with a linear statistical regression between the observations and the mesoscale CAMS product, preliminary downscaled from 10 km to 1 km, due to a geographically weighted regression [53] on NO_x_ and PM emissions downscaled at 1 km resolution [45]. A stochastic residual was computed at observation locations as the difference between the observations and the spatial trend, then interpolated by a linear weighting of the residuals. The corresponding weights are based on a spatial autocorrelation model adjusted to the empirical residuals. It quantifies how close the pollution levels at two different locations are to each other. Regarding PM_2.5_, we used a specific version of multivariate Kriging, the so-called cokriging approach [54], to incorporate information from PM_10_ in the downscaling procedure.

### 2.3. Construction of the Epidemiological Models

Model 1 included the variables mentioned above without considering that a pandemic had started during the observation period. This model has to be considered as a reference one, measuring average conditional associations during the whole period.

Model 2 aimed to suggest possible indirect effects of the pandemic, i.e., whether the conditional association of environmental and sociodemographic variables was changed due to the start of the COVID-19 pandemic. The COVID-19 Stringency Index (CSI) [55] was used to identify when COVID-19-related restriction measures could be assumed to start affecting the daily organisation of society; in particular, the week when the average CSI surpassed the 0.7 cut-off for the first time since the start of the pandemic was chosen. This indicator was included among the predictors and in the interaction with environmental and sociodemographic predictors. A global Chi-square test on the interaction terms was performed, and, in case of lacking statistical significance, the model was repeated without interactions. In cases of statistical significance, separate tests for the environmental and sociodemographic variables were performed, and, again, the model was repeated without the interactions with the block of variables, possibly failing to show statistical significance. The interaction terms investigated whether the COVID-19 pandemic modified the association between each factor and the contact rate with psychiatric services after controlling for the other predictors: the main effects measure conditional associations before the pandemic, while the sum of the main and interaction effects measures the ones during the pandemic.

Model 3 aimed to assess whether different restriction levels (based on the CSI) had a conditional association with the probability of having psychiatric contacts and to investigate the possible differential effect of lockdown and intermediate restriction. Model 3 was based on Model 1, adding variables related to COVID-19 restrictions. The average CSI was calculated for all the weeks from the beginning of the pandemic period (and set to 0 before the pandemic), and the weeks were divided into three groups according to: ‘no or limited restrictions’ (for values up to 0.7), ‘intermediate restrictions’ (for values between 0.7 and 0.8), and ‘lockdown’ (for values above 0.8). The percentage of working days was included in the regression to consider that not all weeks have the same number of working days (for example, due to public holidays) and that when the proportion of working days is higher, the probability of contacts is higher. Furthermore, since the COVID-19 restrictions included restrictions on travel, an indicator variable for weeks that included public holidays on days when inter-regional travel was either banned or discouraged was inserted to take a possible lower reducing effect (due to more health workers being available) of holidays in such weeks into account. Finally, two dummies (for years 2020 and 2021, respectively) were included so that the model could be interpreted as following a ‘difference in differences’ [56] approach: in particular, the difference in the rate of contacts across years in the weeks of lockdown and intermediate restrictions was compared to the same difference in weeks of no or reduced restrictions. This difference was interpreted as the effect of restrictions. 

For all models, global tests were performed for sociodemographic, environmental, and, when present, COVID-19-related restriction variables separately. Only in the case of global statistical significance did we proceed to analyse single predictors. No preliminary selection of predictors was performed; all predictors were included, in order to reduce the risk of omitted-variable bias.

All statistical analysis were performed in Stata 18 [57].

## 3. Results

During the study period, 3923 patients had at least one contact with mental health services of the Verona Department of Mental Health (Table 1); the largest age group was 45–64 years old, 57% of the patients were women, 14% were foreign citizens, almost half of the patients were single, around three-quarters lived with family members, and the most frequent diagnostic group was patients having a neurotic and somatoform disorder (36%), followed by affective disorders (21%), and schizophrenia-spectrum disorders (17%).

Figure 1 compares the trend of CSI with the trend in daily contact rate, highlighting that (at least until the beginning of 2021, when the CSI tended to stabilise) increases in CSI typically coincided with decreases in average contact rates, and vice versa.

In Model 1 (Table 2), both environmental variables (*p* < 0.001) and sociodemographic variables (*p* = 0.002) globally had a significant conditional association with contact rates (results of global tests are reported in Appendix A). Areas with more people living in rented apartments (*p*-value 0.007), lower schooling (*p*-value 0.017), and a higher NO_2_ concentration (*p*-value < 0.001) were associated, ceteris paribus, with an increase in the number of contacts.

In Model 2 (Table 3), global significance was found for sociodemographic and environmental variables separately (*p*-value < 0.001 in both cases). For the pre-pandemic period, tree cover and solar radiation in the previous week were found to be significantly associated with lower contact rates. On the other hand, the rates of inhabitants with, at most, a primary school education and of inhabitants living in rented apartments were found to significantly increase contact rates. As for interaction effects with the pandemic period, statistical significance was met both globally and for sociodemographic and environmental variables separately (*p*-value < 0.001 in all cases). All the interaction terms with the sociodemographic variables were statistically significant, as were the ones with the percentage of tree cover and the presence of green areas. In all cases, the sign of these coefficients was opposite to the one found by the corresponding main term (measuring the estimated conditional association in the pre-pandemic period), suggesting the effect of such predictors on contacts was strongly reduced, eradicated, or (for the presence of green areas, which however did not find significance in its parameter related to the pre-pandemic period) even reversed during the COVID-19 pandemic. Furthermore, solar radiation in the previous week emerged as statistically significant in reducing the probability of contacts.

In Model 3 (Table 4), the global test showed statistical significance only for sociodemographic factors (global significance: *p*-value 0.002) and CSI-related parameters (*p*-value < 0.001), while not for environmental ones (*p*-value 0.630, Appendix A). Higher rates of people living in rented apartments (*p*-value 0.007) and with lower schooling (*p*-value 0.017) increased the rate of contact with psychiatric services. The lockdown (Beta = −1.51) was estimated to significantly reduce contact rates more strongly (*p*-value 0.004) than intermediate restrictions (Beta = −0.54). Figure 2 compares the trends in contacts between patients with and without psychosis, highlighting similarity.

## 4. Discussion

This study investigated the influence of sociodemographic factors and environmental factors in explaining the utilisation of community-based mental health services both before and during the COVID-19 pandemic in Italy. Overall, the influence of certain sociodemographic factors (such as residing in rented apartments and having a lower level of education) and environmental variables (including higher NO_2_ concentration and lower solar radiation) exhibited a consistent pattern, leading to increased utilisation of mental health services over the two and a half-year observation period. However, when examining the pre-pandemic and pandemic periods separately, it becomes apparent that the conditional associations observed before the pandemic changed relevantly during it. Living in a neighbourhood with more trees seems to reduce the contact rate without restrictions, as does living in areas with higher education rates, lower numbers of rented apartments, and lower unemployment rates. However, such differences across census blocks in mental health services utilisation were reduced or modified by lockdowns and restrictions. It is possible that having a green space near the living apartment but not being able to access it due to COVID-19 restrictions could be an additional mental health stressor [58,59]. This evidence should also consider the fact that, on one side, the availability of the services was reduced during the pandemic as many facilities, in particular rehabilitation services and daycare, were closed to reduce contagion, and, on the other side, people were hesitant to visit the services in person during the pandemic [41].

To our knowledge, this is the first study to evaluate the associations between sociodemographic and environmental factors and the utilisation of community-based mental health services during a health emergency scenario.

Green spaces, and trees in particular, are known to improve the quality of the living environment by directly removing air pollutants such as PM_2.5_ and NO_2_ from the atmosphere [60]. It is also known that urban green land cover and blue spaces provide natural scenery and places for restoration from stress and mental fatigue [1,17], representing a key point for public mental health worldwide. Recently, the 3–30–300 rule for urban greening has been proposed to provide citizens with equitable access to nature and its benefits [61]. It states that every home, school, and workplace should have at least three trees in view; there should be no less than a 30% tree canopy in every neighbourhood; and there should be no more than 300 m to the nearest public green space from every residence. This empirical rule was recently evaluated with respect to mental health in Barcelona, highlighting an association with better mental health indicators. Barcelona is also a core location, where several teams in the frame of the EU project URBAG evaluated to what degree green infrastructure can be a source of sustainable cities [62].

Although our dependent variable was the number of contacts with secondary-level mental health services instead of a direct measure of mental health and mental well-being in the community, our results seem to confirm the effects suggested by the literature reported above when considering patients with a severity that needs to be treated by secondary-level services.

### Limitations and Strengths

There are several limitations to our study. The primary methodological constraint pertains to the temporal discrepancy between the collection of data on (1) census blocks (2011), (2) tree cover and green and blue areas (2018), and (3) service utilisation (2019–2021). Solar radiation, and PM_2.5_ and NO_2_ concentrations, being time-varying data, were measured weekly. Furthermore, this study used data sourced from a Psychiatric Case Register (PCR) and integrated environmental information with sociodemographic and clinical data from the registry. This approach is valuable for epidemiological research, and the PCR has proven to be a dependable tool for monitoring and assessing the epidemiology of mental health conditions [63,64,65]. Nonetheless, our study is susceptible to information bias since the PCR data were gathered by various teams of mental health professionals during their routine clinical activities. Furthermore, due to the nature of the geographical data sources, other possibly significant environmental variables, such as traffic levels and noise, were not included as predictors. Finally, our outcome variable, the number of contacts, does not directly measure mental health, but it can be considered a proxy for it although we are aware that the closure of services during the lockdown increased the distance between supply and demand.

Despite these limitations, the strength of this study lies in its spatially explicit analysis of the relationship between mental health and various environmental factors, including stressors like air pollutants and beneficial factors such as the presence of green spaces nearby. Unlike previous research, which often examined these factors separately, this study offers a unique perspective by investigating data from both kinds of factors before and during a pandemic. The study also includes a high-resolution model for air pollution, adding to its originality. The combination of these factors distinguishes this study as a significant contribution to the field.

## 5. Conclusions

In conclusion, our study examined the association between the environment and access to mental health services access for individuals with diagnosed conditions. Future research should explore the broader relationship between environmental factors and the overall well-being of the general population, as the association we investigated may be influenced by various confounding factors specific to the mental health subpopulation. Our study provides information on the complex relationship between the environment and mental health, thereby providing policymakers with insights into potential urban interventions to decrease the burden of mental disorders.

## Figures and Tables

**Figure 1 ijerph-21-00661-f001:**
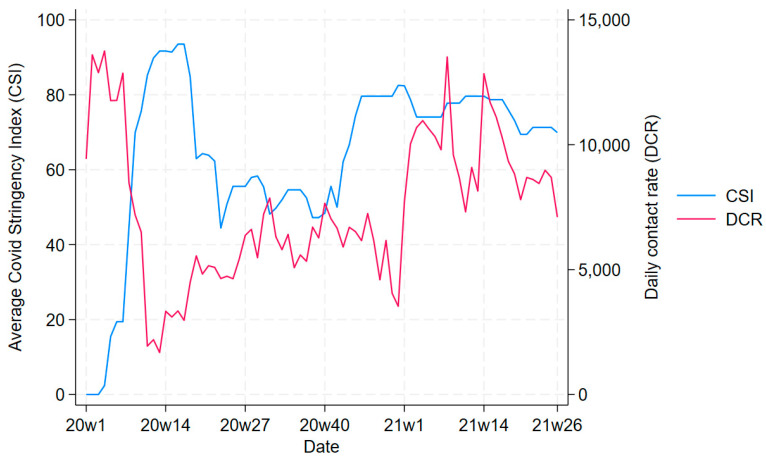
Trends in COVID-19 Stringency Index (CSI) and daily contact rate (DCR) in years 2020 and 2021.

**Figure 2 ijerph-21-00661-f002:**
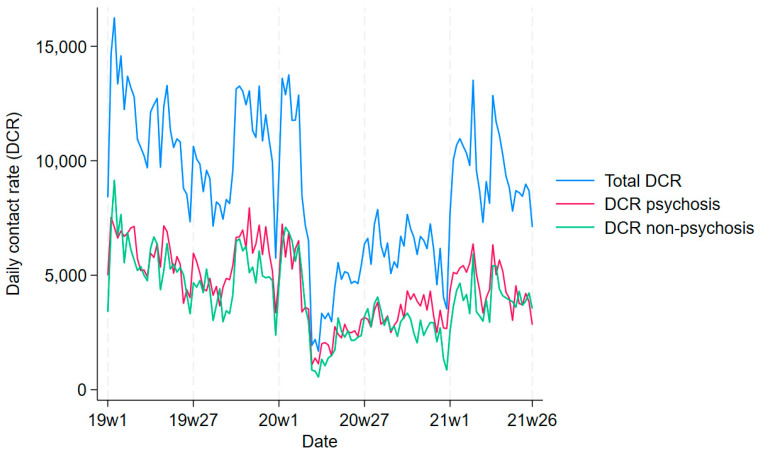
Trends in the global contact rate and the contact rate of patients with and without psychosis in the study period.

**Table 1 ijerph-21-00661-t001:** Sociodemographic and clinical characteristics of patients.

	N	%
All patients	3923	100.00%
Gender		
Female	2225	56.72%
Male	1698	43.28%
Age (n missing = 22)		
18–24 years	312	8.00%
25–44 years	968	24.81%
45–64 years	1736	44.50%
≥65 years	885	22.69%
Citizenship (n missing = 22)		
Italian	3338	85.57%
Others	563	14.43%
Marital status (n missing = 337)		
Single	1684	46.96%
Married	1181	32.93%
Separated/divorced/widowed	721	20.11%
Living situation (n missing = 406)		
Alone	715	20.35%
With family members	2651	75.44%
Sheltered or residential facility	148	4.21%
Diagnosis (n missing = 83)		
Schizophrenia and related disorders	660	16.82%
Affective disorders	806	20.55%
Neurotic and somatoform disorders	1420	36.20%
Personality disorders	290	7.39%
Other diagnoses	664	16.93%

**Table 2 ijerph-21-00661-t002:** Results from Model 1 to predict contact rates without considering information about the COVID-19 pandemic.

	Coefficient	SE	*p*-Value	95% CI
Percentage of working days	−0.746	1.662	0.654	(−4.003; 2.511)
Lagged value of the daily contact rate	**0.089**	**0.0004**	**<0.001**	**(0.088; 0.089)**
Solar radiation (KWs)	−0.170	0.103	0.097	(−0.372; 0.031)
PM_2.5_ concentration	−0.003	0.013	0.824	(−0.027; 0.022)
NO_2_ concentration	**0.081**	**0.016**	**<0.001**	**(0.049; 0.113)**
Percentage of tree cover	−0.036	0.028	0.194	(−0.090; 0.018)
Green areas > 2 hectares around the CB centroid	0.138	0.497	0.780	(−0.836; 1.113)
Watercourses around the CB centroid	−0.194	0.603	0.748	(−1.375; 0.987)
Percentage of inhabitants with at most primary school education	**0.052**	**0.022**	**0.017**	**(0.009; 0.095)**
Percentage of inhabitants living in rented apartments (%)	**0.033**	**0.012**	**0.007**	**(0.009; 0.056)**
Unemployment rate (%)	0.045	0.058	0.435	(−0.069; 0.160)
Lagged value of solar radiation (KWs)	−0.164	0.101	0.103	(−0.362; 0.033)
The proportion of days with PM_2.5_ above the threshold from the previous week	−0.458	0.317	0.148	(−1.079; 0.162)
The proportion of days with NO_2_ above the threshold from the previous week	0.307	0.995	0.758	(−1.643; 2.258)

**Notes.** CI: confidence interval; SE: standard error; KWs: kilowatt hours per square metre; CB: census block; PM_2.5_: particulate matter; NO_2_: nitrogen dioxide. Values associated to significant results are highlighted in bold.

**Table 3 ijerph-21-00661-t003:** Results from Model 2 to predict contact rate including interactions with the COVID-19 pandemic period.

	Coefficient	SE	*p*-Value	95% CI
Percentage of working days	2.430	1.523	0.111	(−0.556; 5.416)
Lagged value of the daily contact rate	**0.088**	**0.0004**	**<0.001**	**(0.088; 0.089)**
Solar radiation (KWs)	−0.138	0.137	0.314	(−0.405; 0.130)
PM_2.5_ concentration	−0.020	0.018	0.248	(−0.055; 0.014)
NO_2_ concentration	0.034	0.019	0.064	(−0.002; 0.071)
Percentage of tree cover	**−0.058**	**0.028**	**0.040**	**(−0.113; −0.003)**
Green areas > 2 hectares around the CB centroid	−0.155	0.506	0.759	(−1.148; 0.837)
Watercourses around the CB centroid	−0.187	0.614	0.761	(−1.391; 1.017)
Rate of inhabitants with at most primary school education	**0.076**	**0.022**	**<0.001**	**(0.032; 0.119)**
Rate of inhabitants living in rented apartments	**0.046**	**0.012**	**<0.001**	**(0.022; 0.070)**
Unemployment rate	0.080	0.059	0.178	(−0.036; 0.196)
Lagged value of solar radiation (KWs)	**−0.424**	**0.139**	**0.002**	**(−0.670; −0.151)**
The proportion of days with PM_2.5_ above the threshold from the previous week	0.175	0.496	0.724	(−0.797; 1.148)
The proportion of days with NO_2_ above the threshold from the previous week	−0.700	1.047	0.504	(−2.753; 1.353)
Pandemic period	−0.269	0.821	0.743	(−1.879; 1.340)
Rate of inhabitants with at most primary school education * pandemic period	**−0.044**	**0.008**	**<0.001**	**(−0.059; −0.029)**
Rate of inhabitants living in rented apartments * pandemic period	**−0.024**	**0.004**	**<0.001**	**(−0.032; −0.016)**
Unemployment rate * pandemic period	**−0.063**	**0.020**	**0.001**	**(−0.102; −0.024)**
Solar radiation (KWs) * pandemic period	−0.163	0.165	0.322	(−0.487; 0.160)
PM_2.5_ concentration * pandemic period	0.032	0.024	0.181	(−0.015; 0.078)
NO_2_ concentration * pandemic period	−0.016	0.025	0.526	(−0.064; 0.033)
Percentage of tree cover * pandemic period	**0.038**	**0.010**	**<0.001**	**(0.020; 0.057)**
Green areas above 2 hectares around the CB centroid * pandemic period	**0.552**	**0.172**	**0.001**	**(0.214; 0.890)**
Lagged value of solar radiation (KWs) * pandemic period	0.243	0.168	0.149	(−0.087; 0.142)
Lagged value of the proportion of days with PM_2.5_ above the threshold from the previous week * pandemic period	−0.659	0.661	0.319	(−1.955; 0.637)
Lagged value of the proportion of days with NO_2_ above the threshold from the previous week * pandemic period	0.817	1.399	0.559	(−1.925; 3.558)
Watercourses around the CB centroid * pandemic period	−0.018	0.216	0.932	(−0.442;0.405)

**Notes.** CI: confidence interval; SE: standard error; KWs: kilowatt hours per square metre; CB: census block; PM_2.5_: particulate matter; NO_2_: nitrogen dioxide. Values associated to significant results are highlighted in bold. The * symbol was used to indicate interactions between variables.

**Table 4 ijerph-21-00661-t004:** Results from Model 3 to predict contact rates with the COVID-19 Stringency Index (CSI) level among predictors.

	Coefficient	SE	*p*-Value	95% CI
Percentage of working days	1.749	1.624	0.282	(−1.435; 4.932)
Lagged value of the daily contact rate	**0.089**	**0.0004**	**<0.001**	**(0.088; 0.089)**
Rate of inhabitants with at most primary school education	**0.052**	**0.022**	**0.017**	**(0.009; 0.095)**
Rate of inhabitants living in rented apartments	**0.033**	**0.012**	**0.007**	**(0.009; 0.056)**
Unemployment rate	0.046	0.058	0.427	(−0.068; 0.161)
Holidays in weeks with a travel ban	0.297	0.312	0.341	(−0.314; 0.908)
Year 2020	**−1.111**	**0.133**	**<0.001**	**(−1.371; −0.851)**
Year 2021	−0.342	0.284	0.229	(−0.900; 0.216)
Lockdown	**−1.510**	**0.286**	**<0.001**	**(−2.070; −0.950)**
Intermediate restrictions	**−0.537**	**0.261**	**0.039**	**(−1.048; −0.026)**

**Notes.** CI: confidence interval; SE: standard error. Values associated to significant results are highlighted in bold.

## Data Availability

The datasets presented in this article are not readily available for privacy reasons. Requests to access the datasets should be directed to Eleonora Prina.

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
