# Peer review of "Relationship between Environmental Conditions and Utilisation of Community-Based Mental Health Care: A Comparative Study before and during the COVID-19 Pandemic in Italy"

_ijerph, 2024, doi:10.3390/ijerph21060661_

Round 1

Reviewer 1 Report

Comments and Suggestions for Authors

I provide here some comments that might improve the quality of the “article”.

Regarding the title, I would recommend some change in the wording, proposing something like:

“Relations between environmental conditions and utilization of community- based mental health care: a comparative study of psychiatric case registration before and during the COVID-19 pandemic in Italy”.

Place the reference in text (line 101) in an appropriate format: “(Kreutz et al., 2023)” and include it in references.

In relation to the objective of the study, it should be talked more in terms of “relationship” (between environmental factors and use of community mental health services) rather than “effect”, which could imply the existence of a “causal relationship” that , neither in this nor in other similar studies could it be proven.

Although the results of the analyzes appear in the supplementary material, it would be of interest if the statistical analysis carried out were detailed in the manuscript (methodology) in more detail, especially regarding the "regression analysis" (method of inclusion/selection of variables in the development of the models -for example: if it was done in steps, etc-).

Reviewer 2 Report

Comments and Suggestions for Authors

: It is very important to assess the impact of environmental conditions, more specifically pollution, on the mental health of people diagnosed with a mental illness. This is also an understudied topic if we take into consideration the magnitude of the problem. Therefore, the authors propose a relevant and much-needed research topic. More specifically, the authors proposed studying the relationship between sociodemographics, environmental conditions, and mental health service utilization, which seems very relevant. The proposition of examining if this relationship changed during COVID-19 seems less clear because service utilization was dramatically affected by lockdowns and mandated decreases in service offers. Also, fear of contamination and leaving home could be important factors that interfere with service utilization during COVID-19. If the outcome variable is the rate of contact with psychiatric and psychological services, how can the artificial decrease in service offered during the pandemic be accounted for? Therefore, I would suggest that the authors bring more clarity about how they were able to account for the artificially decreased service utilization during COVID-19. Perhaps they should consider doing two different analyses – one before COVID-19 and another during COVID-19. The building of three different models to try to take into account the effects of the pandemic made it very difficult to follow the narrative and connect the methodology to the results. Also, of notice, the authors found an association between living in neighborhoods with more trees and reduced contact rates. Could this be due to the fact that higher-income neighborhoods tend to have more trees – therefore, a consequence of socioeconomic status rather than the environment? Finally, the differences between association and causation should be presented more clearly throughout the article.

Comments on the Quality of English Language

Example - the authors wrote:

"Urban nature is found in a recent study to play a central 43 role to creating more equitable, green, livable cities with active inhabitants"

It would read better if they had written:

A recent study found that urban nature plays a central role in creating more equitable, green, livable cities with active inhabitants.

Throughout the article, you can find similar examples.

Round 2

Reviewer 2 Report

Comments and Suggestions for Authors

I'm satisfied with the comments and responses from the authors. I have no further comments or suggestions.